# Image-based phenotyping of disaggregated cells using deep learning

Samuel Berryman[1,2], Kerryn Matthews [1,2], Jeong Hyun Lee[1,2], Simon P. Duffy[1,2,3] & Hongshen Ma [1,2,4,5]✉

The ability to phenotype cells is fundamentally important in biological research and medicine. Current methods rely primarily on fluorescence labeling of specific markers. However, there are many situations where this approach is unavailable or undesirable. Machine learning has been used for image cytometry but has been limited by cell agglomeration and it is currently unclear if this approach can reliably phenotype cells that are difficult to distinguish by the human eye. Here, we show disaggregated single cells can be phenotyped with a high degree of accuracy using low-resolution bright-field and non-specific fluorescence images of the nucleus, cytoplasm, and cytoskeleton. Specifically, we trained a convolutional neural network using automatically segmented images of cells from eight standard cancer cell-lines. These cells could be identified with an average F1-score of 95.3%, tested using separately acquired images. Our results demonstrate the potential to develop an "electronic eye" to phenotype cells directly from microscopy images.

[1] Department of Mechanical Engineering, University of British Columbia, Vancouver, BC, Canada. [2] Centre for Blood Research, University of British Columbia, Vancouver, BC, Canada. [3] British Columbia Institute of Technology, Burnaby, BC, Canada. [4] School of Biomedical Engineering, University of British Columbia, Vancouver, BC, Canada. [5] Vancouver Prostate Centre, Vancouver General Hospital, Vancouver, BC, Canada. ✉email: hongma@mech.ubc.ca

Cancer cell lines have been extensively used to model the disease as well as to screen for potential therapeutic agents. However, since cancer is such a heterogeneous condition, the ultimate utility of these cancer cell line models depends on the ability to accurately classify them. Cancer cell lines are primarily classified based on the histopathology of the original tumor but subtyping of the cell lines may require lengthy molecular and genetic profiling[1–4]. Immunofluorescence phenotyping has contributed to reducing the burden for cell line classification but immunofluorescence relies on the expression of specific cell surface antigens, which is expensive and error-prone, despite efforts in standardizing staining, data collection and automation of analysis[5]. Specifically, immunofluorescence phenotyping may be undesirable because: (1) phenotyping markers may be unavailable or lack specificity, (2) the sample may be too heterogeneous, (3) number of markers required may exceed the number of available fluorescence channels that can be detected, and (4) specific labeling may affect the cell in undesirable ways, such as activation or loss of viability. In many of these situations, an important question is whether individual cells could be phenotyped directly using microscopy images without specific labeling.

Previous approaches for image-based cell phenotyping typically rely on manual feature engineering, which involves extracting specific image features from each cell, such as size, shape, and texture of the nucleus and cytoplasm[6,7]. Machine learning approaches, such as support vector machines or neural networks, are then used to classify cells using these features.

The ability to identify specific phenotypes using this approach is limited because the feature extraction methods must be manually designed, limiting the feature complexity which in turn restricts the ability to distinguish between similar phenotypes. Additionally, feature extraction from microscopy images is a highly variable process that depends on many manually tuned parameters. In order to address these issues, machine learning approaches have been used to phenotype cells directly using microscopy images[8,9]. However, previous phenotyping studies have been restricted to broad cell groups, such as lymphocytes, granulocytes, and erythrocytes, that have morphologies easily distinguishable to the human eye[10–13]. Phenotyping cells with more subtle morphologies has been largely restricted to binary classification in order to detect specific alterations resulting from disease[10,14–17]. Another approach is to use brightfield microscopy images to predict the location of immune-stains on sub-cellular structures in order to identify organelles[18–21]. However, a further step is required to interpret these stains to establish the cell phenotype.

A key challenge in developing machine learning algorithms for classifying cells from microscopy images is segmenting larger microscopy fields into single-cell images. Specifically, adhesion cells are notoriously difficult to segment because they grow next to one another making their boundaries difficult to distinguish[22–24]. This segmentation problem could be dramatically simplified by enzymatically disaggregating cells prior to imaging. However, it is currently unclear if phenotypic information is sufficiently preserved in disaggregated cells.

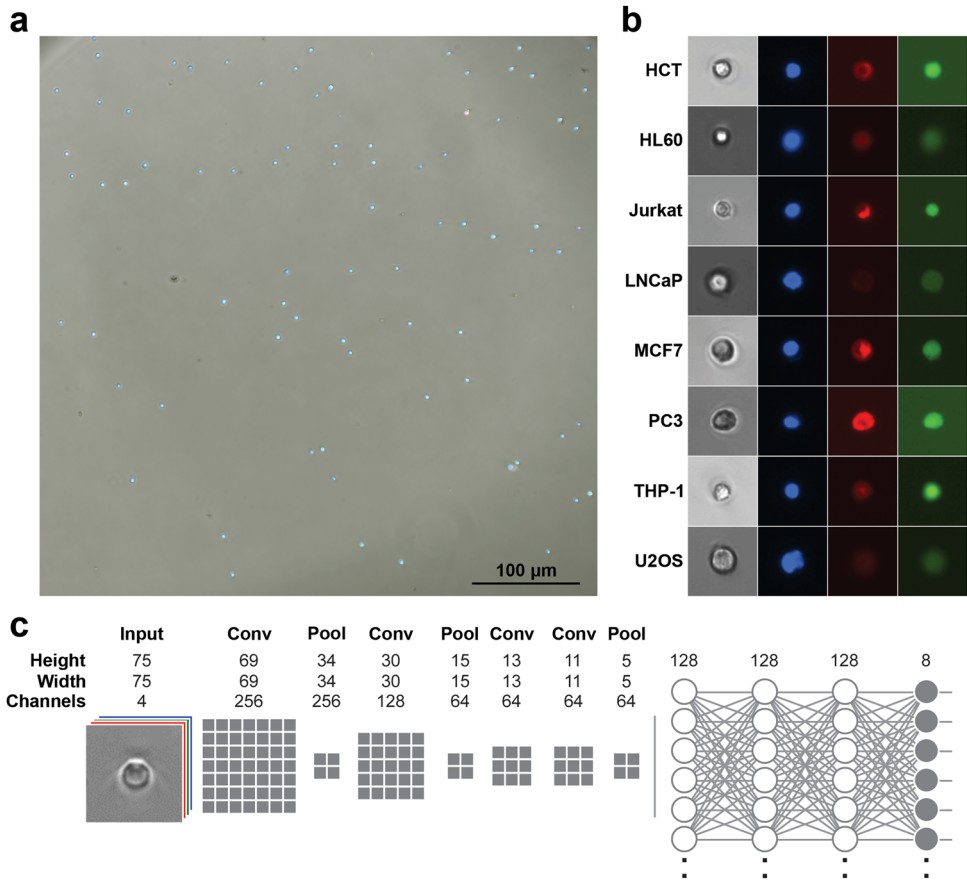

**Fig. 1 Segmentation and network design. a** Sample widefield microscopy image taken on a Nikon TI-2E with a QI2 Camera. The three fluorescent channels are showing nuclear (Hoechst), cytoskeleton (SiR-Actin) and cytoplasm (Calcein). **b** Sample segmented 75 × 75 (55.5 × 55.5 μm) single-cell images. Channels shown from left to right are brightfield, Hoechst, SiR-Actin and Calcein. **c** Network model where the number of grey boxes represents the size of the kernel being used for each convolution layer with a stride of one. Not shown in the image is that after every convolution or fully connected layer is a ReLU activation and batch normalization as well as dropout of 20% after each fully connected layer.

Here, we show disaggregated cells can be phenotyped with a high degree of accuracy from bright-field and non-specifically stained microscopy images using a trained convolutional neural network (CNN). We trained the CNN using $75 \times 75$ pixel brightfield and fluorescence microscopy images of individual cells from eight standard cancer cell lines. All cells were non-specifically stained to identify their nucleus (Hoechst), cytoplasm (Calcein), and cytoskeleton (SiR-actin). The resulting images were indistinguishable to the human-eye, but the CNN was able to classify the cells with a cross-validation accuracy of $96.0 \pm 0.8\%$. We further tested the network using separately prepared cell images and achieved an F1-score of 95.3%. Finally, to investigate the generality of this approach, we imaged cells using a different microscopy system and camera, and then classified the cell images using transfer-learning and achieved an F1-score of 96.0%. This work demonstrates the potential to develop an "electronic eye" to phenotype both adherent and suspension cells based on brightfield and non-specific fluorescence images without the need for specific markers.

## Results

**Imaging**. To acquire a training set for deep-learning, we imaged 8 standard cancer cell lines. These cell lines were derived from colorectal (HCT-116) and prostate (PC3) carcinoma, prostate (LNCaP) and mammary (MCF7) adenocarcinoma, osteosarcoma (U2OS), as well as neutrophilic (HL60), monocytic (THP-1) and T-cell (Jurkat) leukemia. All cells were seeded in 96-well imaging plates and stained using DAPI to visualize the nucleus, Calcein-AM to assess cell viability and cytoplasmic morphology via intracellular esterase activity, and SiR-Actin incorporation to visualize cytoskeletal morphology. Following staining, the cells were imaged using a 10X objective on a Nikon Ti-2E microscope and DS-Qi2 camera, to acquire fields of $2424 \times 2424$ pixels.

**Segmentation**. We developed a Python program to extract $75 \times 75 \times 4$ pixel images of individual cells from wide-field microscopy images. Our program first identified the locations of individual cells by identifying cell nuclei using an Otsu threshold on the DAPI channel (Fig. 1a). Small $75 \times 75 \times 4$ pixel image patches centered on each of the nuclei were then cropped out and fed through a series of tests to reject images containing multiple cells, nuclei, dead cells and debris. First, image patches containing a cell that did not have adequate Calcein-AM present were rejected as non-viable. Second, the watershed algorithm was used on each candidate image patch to determine if multiple objects were present in the nuclear or cytoplasm channels; these were rejected if multiple objects were detected as only single-cell images were desired. Third, any image-patches containing separate fluorescently stained debris, potentially from dead cells, were rejected. Finally, a minimum fluorescence emission threshold (Otsu) was used to ensure each image had the three fluorescent stains present. Together, this digital processing automatically produced a set of high-quality single-cell images with adequate staining from each of the three fluorescent dyes (Fig. 1b). The total number of successfully segmented images for each cell type is listed in Table 1, and sample image patches are shown in Figs. S1–8.

**Network design**. We designed a CNN consisting of a feature extraction and a classification section using the Keras library in TensorFlow. The feature extraction section consists of a series of 4 convolution layers with 3 max-pooling layers. We investigated a range of initial layer kernel sizes, as a larger kernel size is expected to capture more robust features. However, we ultimately selected a kernel size of $7 \times 7$ based on the observation that larger kernel sizes did not greatly improve performance. Each convolution

**Table 1 Size of each class in the training and testing sets.**

| Class | Training | Testing |
|---|---|---|
| HCT-116 | 7151 | 3306 |
| HL60 | 26,298 | 1313 |
| JURKAT | 9208 | 3980 |
| LNCAP | 2555 | 1216 |
| MCF7 | 4109 | 1702 |
| PC3 | 4700 | 1908 |
| THP-1 | 8656 | 1371 |
| U2OS | 15,413 | 1069 |

layer was followed by batch normalization and ReLU activation. The classification section of the network consisted of 3 fully connected layers followed by a smaller fully connected output layer. Each of the 3 fully connected layers was followed by batch normalization, ReLU activation and 20% dropout. The output of the model used a SoftMax error function for backpropagation during training.

**Training**. The training accuracy of a network typically sets a ceiling on the expected outcome of validation accuracy. Therefore, the goal of network design is to reduce the gap between training and validation accuracy. We trained our CNN using a balanced dataset of 10,000 images from each cell phenotype. The dataset was balanced to avoid biasing training results. Classes containing >10,000 cell images were sub-sampled, while classes containing <10,000 cell images were augmented using random integer multiplications of 90-degree rotations. The network was trained over 25 epochs on the dataset. The network was able to train quickly with no volatility due to pairing batch normalization with a small amount of dropout between layers (Fig. 2a). The batch normalization prevented runaway weights and reduced model training time while dropout prevented convergence on local minimums. The final training accuracy for the four-channel model was 98.2%, while the training accuracies were lower for individual channels of brightfield (92.2%), nucleus (91.6%), cytoplasm (95.0%), and cytoskeleton (95.3%).

**Cross-validation**. We initially validated our results using exhaustive five-fold cross-validation. The training-set were randomly divided into five groups. The CNN was trained on four of the groups and tested on the fifth at the end of each epoch. The process was repeated four times, once for each combination. The five-fold cross-validation accuracy for the four-channel model plateaued after 25 epochs with an average accuracy of $96.0 \pm 0.8\%$ (Fig. 2b). The three fluorescent channels achieved similar results with an average accuracy of $85.2 \pm 1.25\%$. The classification accuracy for the brightfield channel was significantly lower at $48.1 \pm 16.0\%$. The accuracy of the four-channel model combined with its low variability suggests a high level of confidence in the classification accuracy of the model.

**Testing**. A key concern in assessing classification accuracy using five-fold cross-validation is the potential for batch effects where sampling artifacts are used by the model for classification[25]. To address this concern, we separately prepared and imaged more cells from the same cell lines, and then classified them using our trained CNN's. Each test set contained 500 randomly sampled images from each class in order to obtain a balanced test set. Using this approach, we found the four-channel model achieved an accuracy of 96.3% (Fig. 2c), which is very similar to the cross-validation accuracy (Fig. 2b). The classification accuracy of the cytoplasm and cytoskeleton channels were 87.3% and 90.2%,

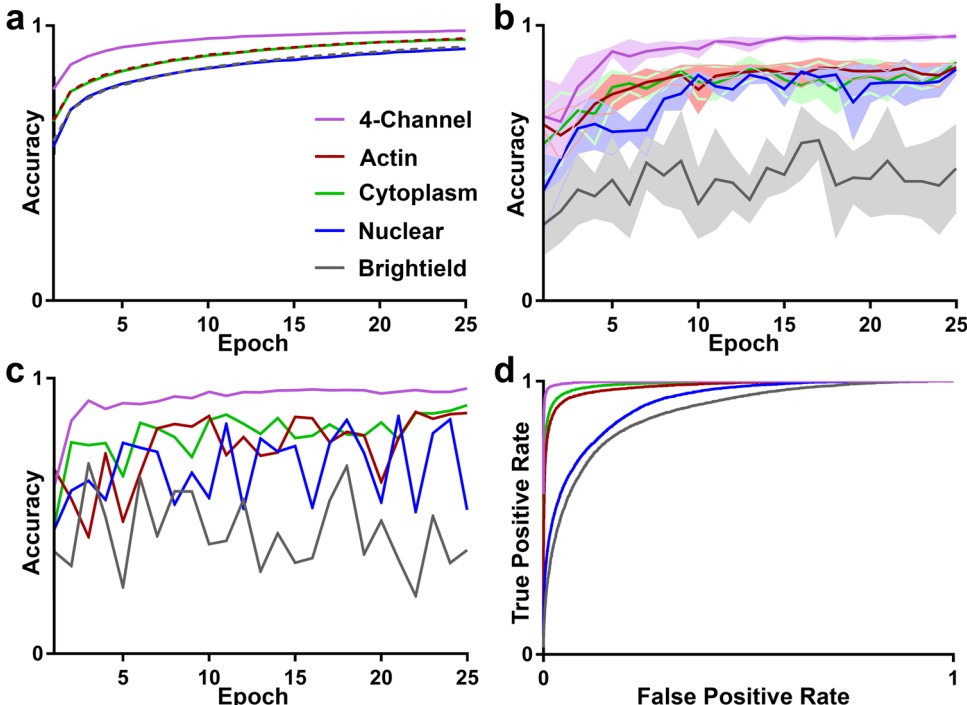

**Fig. 2 Training and Testing. a** Training accuracy results of the five models. Each model was trained over 25 epochs on a balanced dataset of 80,000 images. **b** 5-fold exhaustive cross-validation results of the five models. The four-channel model reached a classification accuracy of 96% with a standard deviation of 0.8% after 25 epochs. **c** Testing results of each of the five models, tested on 500 images of each class, sampled from a new dataset that was separately prepared and imaged. The four-channel model achieved classification accuracy of 96.3%. **d** ROC curve displaying the macro-average trend across classes for each of the five models. The macro-average was computed from the results of each individual classes ROC curve, calculated using a one-vs-all approach.

which is also similar to cross-validation results. The nuclear channel had a much more volatile trend of testing accuracy compared to cross-validation suggesting there may have been some batch effects in the cross-validation results. The brightfield channel performed as expected with a final accuracy of 37.7%.

**Contributions from individual channels**. To investigate the amount of information our method gains by incorporating fluorescent images we trained four networks using either the brightfield or one of the fluorescence channels and compared the classification accuracy with our four-channel model. The results show that the brightfield channel performed the worst out of all channels, implying that the channel had the least relevant information for classification (Fig. 2b–d). We believe this result stems from the brightfield channel only giving insight into the external morphology of cells, which is less distinctive because of disaggregation by trypsin, whereas the fluorescent channels provided insight into the internal structure of the cell. The nuclear channel was found to have the second lowest classification accuracy, which likely stemmed from the lower morphological diversity of nuclei. Finally, the cytoplasmic channel has the greatest single channel classification accuracy, which suggest the greatest amount of phenotypic information in this channel.

**ROC curve**. To investigate how the model would perform in situations were the number of true positive or false positives is of great importance we computed a receiver operating characteristic (ROC) curve (Fig. 2d). The ROC curve was computed by analyzing the output probabilities of the model in a one-vs-all fashion, for each class, then computing the macro-average for each model. The resulting graph shows that the four-channel model had extremely high sensitivity and specificity. The ability

to exchange sensitivity and specificity will be useful for applications, such as rare cell detection, that can tolerate some false positives to capture more true positives. In these results we can see that the cytoplasm and cytoskeleton models are also effective but would have to accept much higher fraction of false positives to achieve a similar yield of true positives.

**Confusion matrix**. To summarize the classification accuracy results for each cell line, we superimposed a confusion matrix on plots of the cumulative distribution for each class and data combination to show the uniformity of the classification probabilities (Fig. 3b). These distribution curves show the confidence of the classification. For example, when the class is predictive of the data, the inference probability function is heavily skewed towards 1 (Fig. 3ai). When the class is unpredictive of the data, the inference probability function is heavily skewed towards 0 (Fig. 3aii). When the class is partially predictive of the data, the distribution function had an in-between shape (Fig. 3aiii). The overlapping confusion matrix shows the resulting number of samples that were classified into each class on the entire unbalanced testing dataset (Fig. 3b). The classification results are summarized in Table 2, reporting the precision, recall and F1-score for each class. The average F1-score was 95.3%, with the best performing class, HCT, having an F1-score of 98.8% and the worst performing class, LNCaP, having a score of 90.5%. The LNCaP's lower score corresponded with it having the lowest number of training samples (2,555 samples).

**Clustering**. We clustered the cell images via t-distributed stochastic neighbor embedding (t-SNE) plots. Using the output of the layer preceding the final output layer, we removed the final layer from the model and recorded the 128-feature output of the

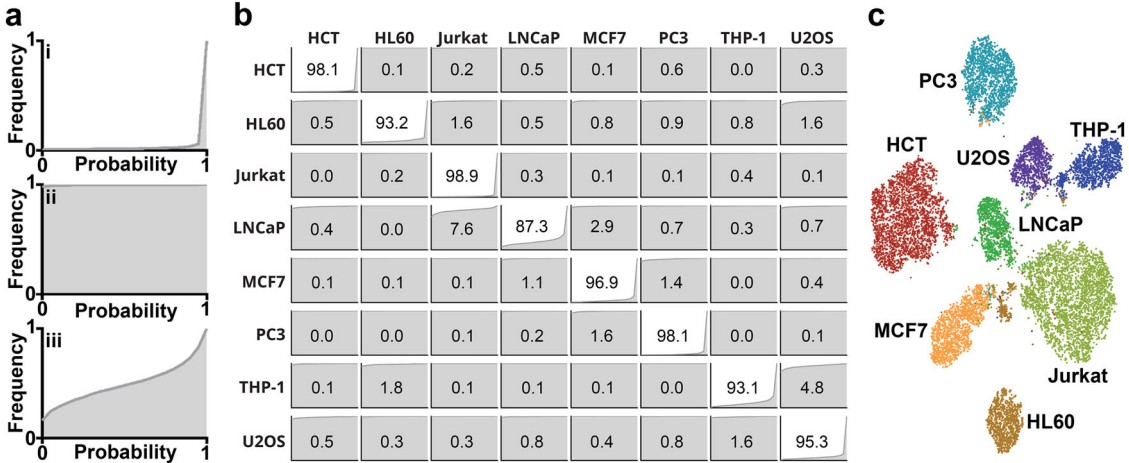

**Fig. 3 Classification accuracy and clustering. a** Example cumulative frequency probability distributions for when the model is predictive (i), unpredictive (ii), and partially predictive (iii) of the class. **b** Confusion matrix of the entire testing dataset. Diagonal values correspond to classification accuracy (recall). Graphs in each position are the cumulative frequencies of classification probabilities for the corresponding classes. **c** t-SNE graph computed from the 128-features extracted from the layer preceding the final classification layer for each image in the testing dataset.

**Table 2 Result metrics from the testing dataset.**

| Class | Precison | Recall | F1-score |
|---|---|---|---|
| HCT-116 | 0.994 | 0.981 | 0.988 |
| HL60 | 0.970 | 0.932 | 0.951 |
| JURKAT | 0.969 | 0.989 | 0.979 |
| LNCAP | 0.940 | 0.873 | 0.905 |
| MCF7 | 0.950 | 0.969 | 0.960 |
| PC3 | 0.960 | 0.981 | 0.970 |
| THP-1 | 0.963 | 0.931 | 0.947 |
| U2OS | 0.896 | 0.953 | 0.924 |
| AVERAGE | 0.955 | 0.951 | 0.953 |

layer. We then used t-SNE with a perplexity of 64 and 2000 iterations, to visualize this higher dimensional data in a two-dimensional graph (Fig. 3c). The graph shows that the model is able to cluster the classes into individual clusters with high separation. This separation is a visualization of the extracted feature quality, demonstrating that sufficient information is not only present but ideal for classification. In addition, the quality of the clusters suggests that classical machine learning approaches could be used to separate the classes, using these learned features, with high confidence.

**Classifying and clustering previously unseen cells.** To investigate how our CNN might respond to previous unseen cell types, we systematically designated a single-cell line as the previously "unseen cell line" and trained the CNN using only the seven other cell lines. We then classified the unseen cell line against these trained cells. This process was repeated by omitting each individual cell line and then classifying it against the remaining cells, allowing us to construct a matrix of cumulative classification probabilities (Fig. 4a). The majority of the resulting probability distributions had the shape of an unpredictive class similar to Fig. 3aii. However, for each class, there were one or two probability distributions functions that were partially predictive, similar to Fig. 3aiii. Interestingly, these partially predictive classes were generally from a cell line related to the queried cell line. For example, HL60 leukemia cells were partially classified as Jurkat and THP-1 cells, which are also leukemia cell lines. Similarly, the epithelial breast cancer cell line, MCF-7 were partially classified as LNCaP and PC3 cells, which are epithelial prostate cancer cell lines. In some cases, the cell lines did not show lineage-specific classification. LNCaP classified with both Jurkat T-cell line as well as MCF-1 mammary adenocarcinoma line. However, this result may reflect the fact that LNCaP had the smallest training set and the lowest classification accuracy during the validation experiments. U2OS also classified with both leukemia and carcinoma cells but this likely reflects the fact that there were no other sarcoma cell lines for comparison.

Using the data generated by querying each cell line against the other seven cell lines, we visualized the clustering of each query cell line using t-SNE plots (Fig. 4b). In this visualization, the queried cell lines formed well-separated clusters from trained cell lines and were generally located near cell lines of a similar lineage. For example, clusters for MCF-7 cells were located near LNCaP and vice versa. Similarly, clusters for HL60, Jurkat, and THP-1 were located near each other. These plots demonstrate that the features learned from the other seven cell-lines have the potential to represent new phenotypes in biologically relevant ways.

**Generality of approach.** To investigate the generality of our approach, we imaged four cell lines using a different microscopy system and then classified them using transfer learning. Specifically, we imaged the cells using a Nikon Ti-E microscope with a QImaging camera and 20× objective. The higher magnification objective approximately compensated for the larger pixel size of the QImaging camera, enabling reuse of the same network architecture. The transfer-learning and testing datasets were acquired following the previous protocol in order to capture the feature and illumination variance in each of the imaging channels (Table 3). The originally learned convolution layers, which were de-activated during training, perform exceptionally well on classifying the images from the new imaging system, achieving final testing accuracies of 96.1% (4-channel), 68.1% (Brightfield), 88.8% (Nuclear), 81.8% (Actin) and 83.5% (Cytoplasm) (Table 4). Importantly, the transfer parameters converged quickly, demonstrating the generalizability of the previously trained convolution layers (Fig. 5a). The confusion matrix also showed slightly higher specificity, while the t-SNE plot shows greater separability of different phenotypes (Fig. 5b, c). These results confirm that the original convolution layers were trained on biologically relevant features for distinguishing different cell phenotypes, demonstrating the generality of our approach for image-based cell phenotyping.

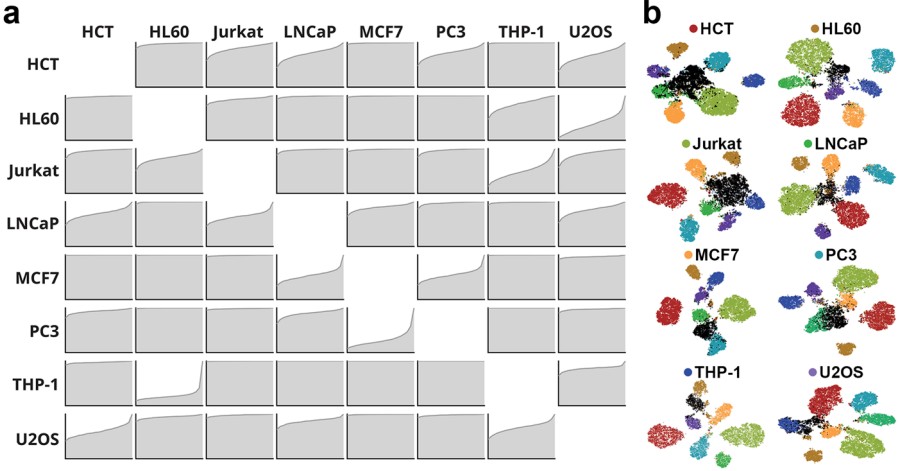

**Fig. 4 Classification and clustering of previously unseen cell lines. a** Classification probability distribution functions for each cell line when it is withheld from training. **b** t-SNE plots of each cell line when it is withheld from training. The withheld class is shown in black, while the trained cell lines are labeled corresponding to Fig. 3c.

**Table 3 Size of each class in the transfer-learning training and testing sets.**

| Class | Training | Testing |
|---|---|---|
| JURKAT | 4661 | 1003 |
| LNCAP | 4123 | 1135 |
| RAJI | 5544 | 1627 |
| THP-1 | 3151 | 1562 |

**Table 4 Result metrics from the transfer-learning testing dataset.**

| Class | Precison | Recall | F1-score |
|---|---|---|---|
| JURKAT | 0.918 | 0.977 | 0.946 |
| LNCAP | 0.985 | 0.974 | 0.979 |
| RAJI | 0.956 | 0.936 | 0.946 |
| THP-1 | 0.973 | 0.962 | 0.967 |
| AVERAGE | 0.958 | 0.962 | 0.960 |

## Discussion

In this study, we investigated whether deep-learning could phenotype single cells directly from microscopy images that are unidentifiable to the human eye. In contrast to traditional immunophenotyping where cell identities are determined using cell-specific antigen profiles, this study employed the distinct strategy of staining cells for common features, and then using deep-learning to distinguish cells from microscopy images. We examined eight cell lines that were dissociated by trypsin and stained with DAPI nuclear stain, Calcein-AM cytoplasmic stain, and SiR-actin cytoskeletal stain. Following unaided segmentation of microscopy images, we developed a CNN model that achieved a five-fold cross validation accuracy of 96.0 ± 0.8%, as well as an average F1-score of 95.3% on a separately acquired testing dataset. To determine whether the CNN model could be used to detect previously unseen cells, we queried each single-cell type against a model generated by the other cell lines. While the efficacy of this approach varied between cell lines, it was remarkable that previously unseen cells were classified as cell lines in related differentiation lineages.

Analyzing microscopy images of trypsin-dissociated cells presented a tremendous challenge for image analysis because the enzymatic activity of trypsin protease causes cells to adopt a common spherical morphology. However, trypsin digestion also provides an important practical advantage, because disaggregated cells can be more evenly dispersed in a microscopy well-plate, which improves the robustness of the segmentation process. Previous studies to discriminate cells based on imaging required seeding cells on a surface, where segmentation is more complex and cell-surface interactions can influence cell morphology[26–28] or by looking directly at histopathology tissue slides[29]. Sirinukunwattana et al.[29] applied a deep-learning approach to classifying cells in histopathology tissue slides and was able to achieve an average F1-score, across 4 classes, of 78.4%. In contrast our work demonstrated a significant improvement as we achieved an average F1-score, across 8 classes, of 95.3%. In addition to the improved F1-score our expected baseline error was higher since we had double the number of classes. Disaggregated cell samples provide a simpler, more rapid, and more uniform imaging condition. Therefore, a key contribution of this work is the finding that images of disaggregated cells contain sufficient morphological information necessary for phenotyping.

A potential major hurdle in robust cell phenotyping using deep-learning is batch errors, which result from CNNs being trained on imaging and processing artifacts that do not reflect the biology of the cell phenotype[25]. In order to minimize batch errors, we generated the training set for each cell line from three separately prepared and imaged samples, to allow our model to capture variations in staining efficiency and illumination. We then ensured that our results were not biased by testing our CNN on a fourth separately prepared and imaged dataset. This further round of testing was important as it avoided the potential biasing in our five-fold cross-validation as each fold was generated by randomly sampling the training set. To evaluate the generality of our approach, we further imaged cells using an alternate microscopy system and camera, and then phenotyped the cells using transfer-learning. These cells were classified with an F1-score of 96.0% across four classes, which confirmed that our CNN was trained on biologically relevant features that could be used to identify each cell phenotype.

Together, this work demonstrates a generalizable deep-learning strategy that can classify cells based on unspecific fluorescence images, rather than specific antigens. This finding is important because immunophenotyping can be an expensive and error-prone process, despite efforts in standardizing staining, data collection, and automation of analysis[5]. We observed that the

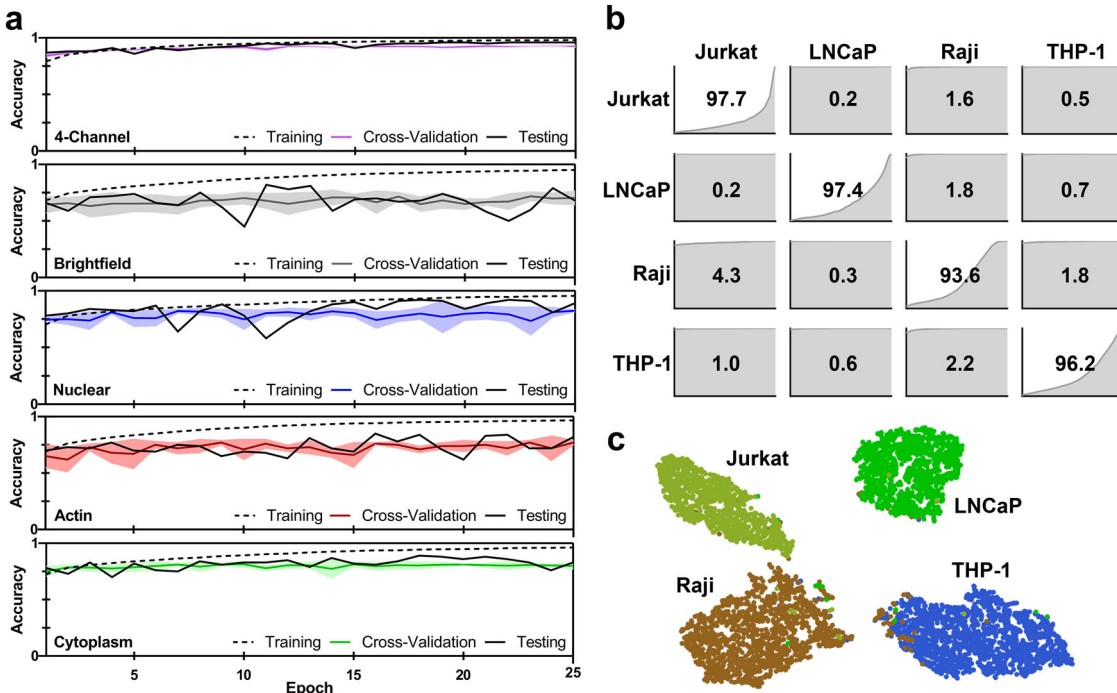

**Fig. 5 Transfer-learning. a** Transfer-learning convergence on images acquired using a different microscopy system. Final testing accuracy were 96.1% (4-channel), 68.1% (Brightfield), 88.8% (Nuclear), 81.8% (Actin), and 83.5% (Cytoplasm). **b** Confusion matrix of the new testing dataset, classified using the four-channel transfer model. Diagonal values correspond to classification accuracy. Graphs in each position are the cumulative frequencies of classification probabilities for the corresponding classes. **c** t-SNE graph computed from the 128-features extracted from the layer preceding the final classification layer for each image in the transfer-learning testing dataset.

morphological differences between cells may not need to be apparent to the naked eye in order to be sufficient for cell classification. Consequently, we have developed an approach for using disaggregated cells, similar to the preparation employed in standard laboratory flow cytometry, to establish rapid and robust image-based cell phenotyping of cells. With this capability, it might be possible to develop a model for the 200+ known cell types in the human body in order to detect previously unknown phenotypes or detect phenotypic shifts that occur because of disease or treatment.

## Methods

**Sample preparation**. A total of 8 cancer cell lines were used in this study. Adherent cell lines were prostate derived cancer cell lines PC3 (ATCC CRL-1435) and LNCaP (ATCC CRL-1740), breast cancer MCF7 (ATCC HTB-22), colon cancer HCT 116 (ATCC CCL-247) and bone osteosarcoma U2OS (ATCC HTB-96). Suspension cell lines were leukemic T-cell lymphoblast Jurkat E6.1 (ATCC TIB-152), acute myeloblastic leukemic HL-60 (ATCC CCL-240) and acute monocytic leukemia THP-1 (ATCC TIB-202). All adherent cells, except LNCaP, were cultured in Dulbecco's Modified Eagle Medium (DMEM) with 4.5 g/L D-Glucose and L-Glutamine (Gibco), emented with 10% Fetal Bovine Serum (FBS, Gibco) and 1X Penicillin/Streptomycin (P/S, Gibco). All other cells were cultured in RPMI Medium 1640 with L-Glutamine, supplemented with 10% FBS and 1X P/S. All cells were incubated in T-75 flasks (Corning) at 37 °C with 5% CO2. When needed, the adherent cell lines were released from the flasks with Trypsin-EDTA (0.25%, Gibco) and then washed twice in complete media twice, prior to resuspension for staining. Suspension cells were also washed twice in complete media. After resuspension, cells were stained with 5 µg/mL Hoechst 33342 (H3570, Invitrogen), 50pM SiR-actin (CY-SC001, Cytoskeleton), and Live Green, 2 µg/mL Calcein AM (C1430, Invitrogen), incubated at 37 °C with 5% CO2 for 1 h and then washed twice in PBS. Cells were resuspended in PBS and aliquoted at low density into Greiner Sensoplate 96-well glass bottom multiwell plates (M4187-16EA, Sigma-Aldrich).

**Microscopy**. Microscopy imaging was performed using a Nikon Ti-2E inverted fluorescence microscope. Images were acquired using a Nikon CFI Plain Fluor 10× objective and a 14-bit Nikon DS-Qi2 CMOS camera. Images were captured using four channels: brightfield with phase contrast, DAPI (Nikon C-FLL LFOV, 392/23

nm excitation, 447/60 nm emission and 409 nm dichroic mirror), mCherry (Nikon C-FLL LFOV, 562/40 nm excitation, 641/75 nm emission and 593 dichroic mirror) and EGFP (Nikon C-FLL LFOV, 466/40 nm excitation, 525/50 nm emission and 495 nm dichroic mirror). Illumination for brightfield imaging was performed using the built in Ti-2E LED. Epifluorescence excitation was performed using a 130 W mercury lamp (Nikon C-HGFI). Gain, exposure and vertical offset were automatically determined using built-in NIS functions to avoid user biasing. Cells were imaged in Greiner Sensoplate 96-well glass bottom multiwell plates (M4187-16EA, Sigma-Aldrich). The concentration of cells in each well were diluted down to ~1000 cells to ensure adequate spacing between adjacent cells. An automated procedure was run on NIS using the Jobs function to take 16 images, on each of the 4 channels, inside of each well. The images were exported from NIS to standard TIFF format.

**Segmentation**. The TIFF files were segmented using a custom python script. The script begins by extracting cell locations using a global Otsu-threshold on the DAPI channel (nuclear stain) followed by object labeling using the SciPy library. Images from each location are then checked for usability. Specifically, a 75 × 75 pixel bounding box is defined around the centroid of each detected object. The 75-pixel size was selected because it was sufficiently large to fit single cells from the cell lines imaged using a 10× objective. The proposed image patches are then put through a number of rejection tests. First, the nuclear channel is thresholded using an Otsu threshold on the patch and the number of nuclei present is quantified. If numerous nuclei are detected, or found on the edge of the bounding box, the patch is rejected. The next test involves checking cell viability and counting cell bodies in the cytoplasm channel. This is done by running an Otsu-threshold on the cytoplasm channel and then ensuring a minimal count of pixels is present (50 pixels). If an adequate number of pixels is not present, the cell is assumed to be dead. Object detection is then done using the watershed algorithm on the combination of the three fluorescent channels; if more than one cell body is present, or if the cell body is touching the bounding box, it is rejected. The nuclear and cytoplasm channels were than individually checked likewise for multiple bodies and rejected if multiple objects, or objects on the bounding box, were found. The actin channel was checked only for contact with the bounding box as it was possible for actin filaments to be non-connected in numerous areas of the cell body. Images from acceptable locations were normalized, between 0 and 1, and added to a list of images in a numerical array. The numerical array of images was serialized and saved in pickle format, which has a lossless compression. This process was automatically repeated for each stack of TIFF images in the database.

**CNN model**. A CNN, shown in Fig. 1c, was designed in Python using the Keras library in TensorFlow. The network was designed to accept a 4-channel input of size $75 \times 75$ pixels. The model started with a 256-channel convolution layer with a kernel size of $7 \times 7$ with a stride of 1. The next layer is a $2 \times 2$ max-pooling layer with a stride of 1. Next is a 128-channel convolution layer with a kernel size of $5 \times 5$ and a stride of 1 followed by another $2 \times 2$ max-pooling layer. After this is two 64-channel convolution layers in series, each with a kernel size of $3 \times 3$ and a stride of 1. These layers are followed by a max-pooling layer of $2 \times 2$ and stride of 1 before being flattened for connecting to the fully connected layers. Each convolution layer was followed by batch normalization and ReLU activation. Three 128-node, fully connected layers, each with ReLU activation batch normalization and 20% dropout, were used to learn on the features extracted using the earlier convolutional layers. The output of the network consisted of eight nodes, one for each class, with a soft-max error function used for back propagation. Data augmentation, in the form of random integer multiplications of 90-degree rotations, was used to up-sample the classes that had under 10,000 samples for training to ensure even classes and non-biasing during training.

**Training environment**. The software was run on a single computer operating Windows 10 with an Intel i7-8700K running at 3.70 GHz. There was 64 GB of DDR4 RAM running at 3200 MHz. The graphics card was an 8 GB GTX 1080. Training was done in Python 3.6.8 utilizing the TensorFlow 1.13.1 library.

**Training**. Each iteration of the network was trained on 25 epochs, with Adam optimization and a learning rate of 0.001. The soft-max function was used as an error function for the backpropagation. Several different networks were trained to show the effects of varying different parameters on the inputs. First, the dataset was sub-sampled through cross-validation to investigate the variability of classification accuracy as shown in Fig. 2b. Second, the input layer was modified to accept each individual image channel to quantify the dependency of training on each channel as shown in all of Fig. 2.

**Cross-validation**. To ensure the validation accuracy was accurate, five-fold exhaustive cross validation was used. The cross validation was done by splitting the training-set into five groups. The network was then trained five times on training data consisting of four of the five groups. The fifth group in each training session was used to determine a validation score. The five validation scores were averaged at each epoch to compute the reported validation accuracies.

**Confusion matrix**. To identify the miss-classifications between labels a confusion matrix was constructed. All of the images in the testing dataset were used. A class label was predicted using the CNN and compared against the true class label. The diagonal positions correspond to true labels, while the off-diagonal positions show miss-classifications. The matrix was normalized by the number of images in each class to represent the probabilistic accuracy.

**t-SNE visualization**. To visualize the relative clusters in our data a t-SNE plot was constructed on all of the images in the testing dataset, using the second-to-last fully connected layer of the network. The layer consisted of 128 nodes, giving a 128-feature vector for each validation image. This required removing the final layer from the network, after training, and classifying all the testing images.

**ROC curve**. To validate the sensitivity of the model a ROC curve was calculated on all of the images in the testing dataset, using the probability outputs of each class. The ROC curve was calculated for each class using the SciKit library, which required a one-vs-all classification method in which all the other class are treated as a single group being compared to the current class of interest. As the classes preformed similarly, the macro-average ROC curve was calculated and displayed. The ROC curves were summed together and divided by the number of classes to find the average accuracy.

**Transfer-learning**. The model can be made generalizable to other hardware setups through the use of transfer learning. For our transfer, we utilized a Nikon Ti-E Eclipse inverted fluorescence microscope. Images were acquired using a Nikon CFI Plain Fluor 20× objective and a 21-bit QImaging QIClick Camera. Illumination for brightfield imaging was performed using a 100 W halogen lamp (Ti-Dh Dia Pillar Illuminator). Epifluorescence excitation was performed using a 130 W mercury lamp (Nikon C-HGFI). None of the hardware used in the transfer learning microscope had been previously used in this study. The originally trained models were used as a starting point. The parameters in the convolution layers were fixed, making only the fully connected layers trainable. The final classification layer was replaced with a four-node layer, representative of the four classes (Jurkat, LNCaP, Raji, and THP-1) in the transfer-learning study. These four classes were T-cell lymphoblast Jurkat E6.1 (ATCC TIB-152), prostate derived LNCaP (ATCC CRL-1740), B-cell lymphoma Raji (ATCC CCL-86), and acute monocytic leukemia THP-1 (ATCC TIB-202). Sample preparation and training hyper-parameters were kept the same as previously outlined.

**Statistics and reproducibility**. All replicate experiments are presented as a mean ± standard deviation. The statistical analysis, and plotting, was completed in GraphPad Prism. The T-SNE function from the scikit library was used to transform the classification feature space into two dimensions. Cumulative histograms were created in GraphPad Prism with bin sizes of 0.05.

## Data availability

All imaging data that supports the findings of this study have been made available in .tif format at https://dataverse.scholarsportal.info/dataset.xhtml?persistentId=doi:10.5683/SP2/TDULMF[30].

## Code availability

Custom software and the NIS JOB protocol have been made available at https://github.com/SamBerryman/Image-based-Cell-Phenotyping-Using-Deep-Learning[31].

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

## Author contributions

S.B. and H.M. designed the study; S.B. and J.H.L developed the imaging protocol; S.B. and K.M. performed the experimental work; S.B. developed the software, analyzed results, and prepared figures; S.B., K.M. S.D., and H.M. wrote the article; H.M. supervised the study.

## Competing interests

The authors declare no competing interests.
