## [Peer Review File · Communications Biology]

Reviewers' comments:

Reviewer #1 (Remarks to the Author):

In this paper a deep learning approach is taken to classify bright field and general fluorescence images of cells. 8 standard cancer cell lines are imaged, using multiple fluorescent dyes.

Custom software is written to crop individual cell images from the full size fields, along with a simple pipeline to reject problematic images. Once trained, a number of interesting methods are used to evaluate accuracy of both a combined model and models using individual fluorescent channels. Whilst this evaluation is a strong point, I have some issues with the paper as a whole, as it currently stands.

Comments

1. In some sense the pre-processing pipeline to remove "low quality" images removes much of the challenge potentially faced by the AI system. By ensuring all images used in the system are "high quality" you are making the AI system appear much more capable than in reality it is. I therefore feel it is important to report accuracies for the full cropped dataset, rather than the pre-filtered one, as an additional accuracy measure which is able to gauge realistic performance for real data.

2. The network design here is simple, and certainly not novel in a meaningful sense. A standard classification architecture is used. The pre-processing pipeline may be novel, but it is not a significant contribution, using simple and well understood image processing methods. Still, it would be nice to see some results illustrating the effectiveness of this pipeline, as none are given at present. Is it throwing away all images with particular features? How effective is it? What kind of images get filtered out? This needs more exploration.

3. Please explain the motivation behind your 7 versus 1 classification, where you omit one category at a time from the classification. What does this analysis tell you? The fact that "new" cell lines are classified as "similar" kinds of cells which the system WAS trained on is both unsurprising and worrying. Does this illustrate that a live system would likely miss-classify cells it has not seen before, rather than identify them as "new"? This needs some explanation.

4. I find the cumulative probability graphs in Figs 3 and 4 confusing. Please explain more clearly what they are showing- I am unfamiliar with confusion results presented in this manner. Also, what does the clustering display reveal to a reader? I am not sure.

5. The claim that the cell images are "indistinguishable to the human eye" is a bold one. Such a claim should be backed up by evidence that a suitable trained expert can in fact not distinguish between the image types. From Fig 1b alone it is impossible to verify this claim.

6. In the conclusion, jumping from the 8 tested cell types to an assumption that a system may work well with 200+ is rather a large leap.

7. Is 95% accuracy good enough in this domain? Accuracies above 90% look impressive but are often easy to achieve in classification studies. Is a 1 in 20 error acceptable?

8. Are the software/models/data available on a suitable, long-term open source distribution platform (note, not Google drive as this is unlikely to be available long term)

Minor comments:

- "500 randomly sample*d* images"

- Clustering paragraph: Do you mean Fig 3c?

-p8 Otsu, not Otto

- I Fig 3a missing?

Reviewer #2 (Remarks to the Author):

This paper demonstrates the classification power of a relatively small convolutional network to differentiate cell lines from disaggregated cells. Unlike adherent cultured cells, the cells in the study were treated using trypsin and are then sparsely localized in the well but present the same rounded morphology.

The study was designed by imaging 8 commercially available cell lines in an High Content Screening system cells stained for nuclei, actin and viability checker. After training on cropped view around unique cells, the network demonstrated a good accuracy in cell classification, with the main contribution of actin and viability intensity. Different questions were addressed already in the manuscript, such as contributions of the different channels or experimental design to avoid imaging settings to be influential since only one cell line was present on each images.

0. It would have been interesting to test or comment other classical stains such as other nuclei staining, or using a different microscope with the same magnification, or at least to comment on the transposability : do the author expect to apply Learning transfer, use the same network with a limited training set, retrain integrally?

1. My major question is the interest of using imaging for disaggregated cells. Since the experiment regards disaggregated cells, the interest of spatial organization of different cell types is lost. I would have expected to test or at least report from the literature the accuracy of other cell differentiation methods such as multiplexing /barcoding/ single cell flow sequencing/ flow cytometry and a discussion regarding the advantages in cost and accessibility/speed/ accuracy/ amount of information against these methods.

Minor comments:

2. The authors state "The ability to identify specific phenotypes using this approach [ad hoc image features by known filters] is limited because the features are defined arbitrarily and not necessarily biologically relevant."

Is not it the contrary? manually defined convolutional filters features can be biologically relevant and are clearly easier to interpret than convolutional layers? Assymetry, texture homogeneity, can be mathematically defined and interpreted.

In particular, in the discussion some comment about cell types misclassified quite systematically (even if at low rate) would gain from trying to identify features in common.

3. Construction of the database for training, validation and testing:

the methods is only briefly described in methods for constructing the database of cropped images, including exclusion of data. The source code is provided but should be then commented. It sounds like imaging artefact, polynuclei or other exclusion parameters are detected by watershedting the nuclei mask and counting the number of objects. The minimum emission threshold is not defined in the methods part or the definition of negative signal for Calcein-AM.

Please refer to the step 1 of your code for further details, or at least provide further details.

4. Data should not be hosted on gdrive with an unconvenient 74Gbytes zipped files. Please consider a deposition on public archive that would also give you a doi (zenodo or other). Code should be hosted on GitHub or other system versionning system.

5. References: Ref Cell : In silico Labeling not properly cited (science direct link, no authors etc...). Set the doi for all references please.

6. Fig 1 : more useful for the reader would be to have the cell line types names indicated to demonstrate the undistinguished cell features stated.

7. Fig 2A: non readable as curve are superimposed. Consider different line plot (dotted, dashed...)

8. scale bar are missing from all images, and no indication regarding pixel size is given in the text.

Dear Editors and Referees,

Please find enclosed our revised manuscript, “Image-based Cell Phenotyping Using Deep Learning” for consideration for publication in Communications Biology.

We thank the reviewers for their comments and their encouragement. We have addressed their comments in a substantially revised manuscript with additional experimental results to support our findings.

Reviewer #1:

- 1. In some sense the pre-processing pipeline to remove "low quality" images removes much of the challenge potentially faced by the AI system. By ensuring all images used in the system are "high quality" you are making the AI system appear much more capable than in reality it is. I therefore feel it is important to report accuracies for the full cropped dataset, rather than the pre-filtered one, as an additional accuracy measure which is able to gauge realistic performance for real data.**

We thank the reviewer for expressing this concern. Our pre-processing pipeline does not remove low quality images. It removes images of debris, cell clusters, and non-viable cells. This process does not distinguish between good cells and bad cells, but only extracts images single viable cell. Moreover, our procedure for pre-processing the imaging data is similar to the process used to generate other standardized annotated datasets. We clarified the description of our pre-processing pipeline on lines 99-110, 350-352 and 358-364. Additionally, we provided images of cells accepted and rejected by the pre-processing pipeline in Fig. S1-S11 in the supplemental materials.

- 2. The network design here is simple, and certainly not novel in a meaningful sense. A standard classification architecture is used. The pre-processing pipeline may be novel, but it is not a significant contribution, using simple and well understood image processing methods. Still, it would be nice to see some results illustrating the effectiveness of this pipeline, as none are given at present. Is it throwing away all images with particular features? How effective is it? What kind of images get filtered out? This needs more exploration.**

We thank the reviewer for this comment. The novelty of our work is not the network design, but rather the application of convolutional neural networks to phenotype disaggregated cells. We intentionally designed our network using minimal complexity to achieve the goal of phenotyping disaggregated cells. The high classification accuracy achieved using this network confirms that sufficient information is present in the images of disaggregated cells to correctly identify their phenotype. As mentioned in the previous comment, the pre-processing pipeline was not designed to throw away unwanted samples but rather to

determine if a single viable cell was present in each image. We clarified these issues on lines 99-110, 350-352 and 358-364.

- 3. Please explain the motivation behind your 7 versus 1 classification, where you omit one category at a time from the classification. What does this analysis tell you? The fact that "new" cell lines are classified as "similar" kinds of cells which the system WAS trained on is both unsurprising and worrying. Does this illustrate that a live system would likely miss-classify cells it has not seen before, rather than identify them as "new"? This needs some explanation.**

The motivation for the 7 vs 1 classification is to characterize how the network would behave when classifying a new (i.e. previously unknown) cell type. What we learned from these tests is that classifying previous untrained cells results some non-negligible classification probabilities towards biologically related cell types. This result tells us the network has learned biologically relevant features that could potentially be used to identify new cell types, as well as to find their closest relatives. Additionally, the classification probabilities for previously unknown cell types are significantly lower, which suggests it may be possible to design heuristics for identifying new cell types. We clarified these points in lines 220-226.

- 4. I find the cumulative probability graphs in Figs 3 and 4 confusing. Please explain more clearly what they are showing- I am unfamiliar with confusion results presented in this manner. Also, what does the clustering display reveal to a reader? I am not sure.**

We thank the reviewer for pointing out this issue and we revised Fig. 3 (provided below) to better explain the value of the cumulative probability distribution. The cumulative probability graphs are used to visualize the distribution of the inference probabilities on each class and data set combination. When the model is predictive of the class, the inference probabilities are distributed near 1 (Fig. 3a(i)). When the model is not predictive of the class, the inference probabilities are distributed near 0 (Fig. 3a(ii)). When the model is partially predictive of the class, the inference probabilities are in between the previous two cases (Fig. 3a(iii)). When the model is asked to infer a previously unseen cell type, the inference probability distribution may be similar to Fig. 3a(ii) for unrelated cell types, or similar to Fig. 3a(iii) for related cell types. Clustering shows the relationship between different cell types. Closely related cell types are clustered closer together. We have clarified these ideas on lines 191-201, 207-208, 209-212 and 237-238.

Figure 3 | Classification accuracy and clustering. a, Example cumulative frequency probability distributions for when the model is predictive (**i**), unproductive (**ii**), and partially predictive (**iii**) of the class. **b**, Confusion matrix of the entire testing dataset. Diagonal values correspond to classification accuracy (recall). Graphs in each position are the cumulative frequencies of classification probabilities for the corresponding classes. **c**, t-SNE graph computed from the 128-features extracted from the layer preceding the final classification layer for each image in the testing dataset.

5. The claim that the cell images are "indistinguishable to the human eye" is a bold one. Such a claim should be backed up by evidence that a suitable trained expert can in fact not distinguish between the image types. From Fig 1b alone it is impossible to verify this claim.

We thank the reviewer for this comment. In response, we have provided significantly more example images of each cell type as supplemental information in Fig. S1-8. As shown in these figures, cells in their disaggregated form have very limited external morphology. Therefore, it is quite obvious that human cognition will not be able to identify the cell type with or without training. Furthermore, there are no trained experts for this particular task. We have tempered our language on this issue on lines 21-23, 27-29 and 259-260.

6. In the conclusion, jumping from the 8 tested cell types to an assumption that a system may work well with 200+ is rather a large leap.

We thank the reviewer for this comment. We agree that jumping from 8 to 200+ classes is a big leap. This statement is aspirational, and we do not suggest that our network have been trained for this task. However, other work classification on image classification using deep learning has been shown to scale well to larger number of classes. Therefore, we hope such ideas will inspire subsequent work to scale up this approach to study the human cell atlas. We have tempered our language on this issue on lines 306-308.

7. **Is 95% accuracy good enough in this domain? Accuracies above 90% look impressive but are often easy to achieve in classification studies. Is a 1 in 20 error acceptable?**

Yes, we believe 95% accuracy is sufficient. Cell phenotyping using flow cytometry typically have errors >5%.

8. **8. Are the software/models/data available on a suitable, long-term open source distribution platform (note, not Google drive as this is unlikely to be available long term)**

Custom software have been made available at <https://github.com/SamBerryman/Image-based-Cell-Phenotyping-Using-Deep-Learning>. All imaging data that supports the findings of this study have also been made available at <https://dataverse.scholarsportal.info/privateurl.xhtml?token=69fa58d9-8e9e-43f0-ab20-39c35429925d>.

Minor comments

1. **"500 randomly sample*d* images"**

Fixed.

2. **Clustering paragraph: Do you mean Fig 3c?**

Fixed.

3. **p8 Otsu, not Otto**

Fixed.

4. **I Fig 3a missing?**

?

Reviewer #2:

1. **It would have been interesting to test or comment other classical stains such as other nuclei staining, or using a different microscope with the same magnification, or at least to comment on the transposability : do the aithor expect to appluy Learning transfer, use the same network with a limited training set, retrain integrally?**

We thank the reviewer for this comment, and we agree that the transposability of our network via transfer learning is important to demonstrate the generalizability of our

approach. To address this comment, we imaged more cells using a different microscope (Nikon Ti-E) with a different objective magnification (20X) and camera (QImaging). We then used transfer learning to classified these cells, and achieved a F1-score of 96.0% with rapid convergence. Changes to the manuscript related to this work can be found in lines 242-255, 437-449, Table 3, Table 4, and Figure 5 (provided below).

Figure 5 | Transfer-Learning. **a**, Transfer-learning convergence on images acquired using a different microscopy system. Final testing accuracy were 96.1% (4-channel), 68.1% (Brightfield), 88.8% (Nuclear), 81.8% (Actin) and 83.5% (Cytoplasm). **b**, Confusion matrix of the new testing dataset, classified using the 4-channel transfer model. Diagonal values correspond to classification accuracy. Graphs in each position are the cumulative frequencies of classification probabilities for the corresponding classes. **c**, t-SNE graph computed from the 128-features extracted from the layer preceding the final classification layer for each image in the transfer-learning testing dataset.

2. My major question is the interest of using imaging for disaggregated cells. Since the experiment regards disaggregated cells, the interest of spatial organization of different cell types is lost. I would have expected to test or at least report from the literature the accuracy of other cell differentiation methods such as multiplexing /barcoding/ single cell flow sequencing/ flow cytometry and a discussion regarding the advantages in cost and accessibility/speed/ accuracy/ amount of information against these methods.

This question is precisely the reason why we performed this study – we wanted to see if it is possible to phenotype cells from imaging without the spatial distribution of cellular components associated with attachment to a surface. This ability to phenotype disaggregated cells could dramatically simplify and expediate phenotyping by obviating

the need to attach cells to a surface or to immuno-stain for specific markers. There are limited previous work available on phenotyping and no previous work on phenotyping using disaggregated cells. In one related work using deep-learning to phenotype cells from histopathology slides into 4 types obtained an average F1-score of 78.4%. Our study phenotyped cells into 8 classes and achieved an F1-score of 95.3%. We discussed this issue and compared our work with others on lines 276-278. Additionally, we reported F1 scores in Table 2.

Minor Comments

- 1. The authors state "The ability to identify specific phenotypes using this approach [ad hoc image features by known filters] is limited because the features are defined arbitrarily and not necessarily biologically relevant." Is not it the contrary? manually defined convolutive filters features can be biologically relevant and are clearly easier to interpret than convolutional layers? Assymetry, texture homogeneity, can be mathematically defined and interpreted. In particular, in the discussion some comment about cell types misclassified quite systematically (even if at low rate) would gain from trying to identify features in common.**

We agree that classification using manually defined features may give results that are easier to interpret for humans. However, our point is that manually defined features are limited by human cognition, which may not be able to phenotype cells from non-specific fluorescence images. Therefore, classification of microscopy images directly using convolutional neural networks provides greater possibility of finding distinguishing features of particular cell phenotypes. We discussed these issues further on lines 49-52.

- 2. Construction of the database for training, validation and testing: the methods is only briefly described in methods for constructing the database of cropped images, including exclusion of data. The source code is provided but should be then commeted. It sounds like imaging artefact, polynuclei or other exclusion parameters are detected by watershedting the nuclei mask and counting the number of objects. The minimum emission threshold is not defined in the methods part or the definition of negative signal for Calcein-AM. Please refer to the step 1 of your code for further details, or at least provide further details.**

We thank the reviewer for these suggestions and provide responses below:

- We revised our methods to provide more details. Specifically, the watershed algorithm was used on the nuclear channel to initially determine the potential locations of cells. Next, all three of the fluorescent channels are analyzed to determine if the potential position should be added to the dataset. These tests verify

that the image contains: (1) a viable cell, (2) a single cell and (3) the detected cell does not lie on the image boundary. We clarified these issues further on lines 350-352 and 358-364 as well as in our code.

- We provided additional comments in our code.
- We are assuming the question on minimum emission threshold is referring to the determination of cell viability. The minimum threshold is the calculated Otsu-threshold and the minimum pixel count is 50. This gives us a mean-diameter of 5.2 μm , significantly smaller than our expected cell diameter. We clarified this issue on lines 358-364.

- 3. Data should not be hosted on gdrive with an unconvenient 74Gbyres zipped files. Please consider a deposition on public archive that would also give you a doi (zenodo or other). Code should be hosted on GitHub or other system versionning system.**

Agreed. Custom software have been made available at <https://github.com/SamBerryman/Image-based-Cell-Phenotyping-Using-Deep-Learning>. All imaging data that supports the findings of this study have also been made available at <https://dataverse.scholarsportal.info/privateurl.xhtml?token=69fa58d9-8e9e-43f0-ab20-39c35429925d>.

- 4. References: Ref Cell: In silico Labeling not properly cited (science direct link, no authors etc...). Set the doi for all references please.**

References have been fixed and doi added.

- 5. Fig 1: more useful for the reader would be to have the cell line types names indicated to demonstrate the undistinguished cell features stated.**

Cell line names have been added to **Fig. 1**. Additional images have also been made available in **Fig. S 1-8**.

- 6. Fig 2A: non readable as curve are superimposed. Consider different line plot (dotted, dashed...)**

Modified **Fig. 2a** so that the overlapping lines are dashed.

- 7. scale bar are missing from all images, and no indication regarding pixel size is given in the text.**

We have added a scale bar to **Fig. 1a** and added a description for the image patch scale in the caption.

REVIEWERS' COMMENTS:

Reviewer #1 (Remarks to the Author):

Thank you for taking the time to improve the paper based on my comments. Most of these are well handled, I just have a few follow up comments on the below:

Item 5: Thank you for tempering the language and adding more example images. However, I still think the claim they are indistinguishable by eye is too strong. Looking at S1 to S8, I agree some are very similar, but some are "easy" to differentiate even for me (not a trained expert in the field). For example, S1 seems distinctive, as do S7 and S8. Please could the authors further re word the claims, or better justify them.

Typos:

L359 thresholde*d*

L237 "trained cell lines *and?* were generally"

Reviewer #2 (Remarks to the Author):

Automated cell classification is an important yet a challenging computer vision task. In recent years, there have been several studies attempting to build an artificial intelligence-based cell classifier using cellular images obtained from an optical microscope, usually based on a set of human defined features. Some are attempting to do it label-free, while here the authors are taking advantage of 3 classical staining in addition to the phase contrast imaging. Also one of the main difference in the approach is that here the authors ask the question of the possibility of cell line identification from disaggregated cell using widely available inversed fluorescence microscopes, relieving some of the difficulties of single cell segmentation in adherent cells, but then losing usual morphological information.

Compared to the previous version, authors have made the effort of sharing the dataset on dataverse and their code. One comment for reuse is that the true cell class seems to be given by the file name but this could be specified in the text to help reproducing the results.

They also have added experiments about the transposability by using a different microscope with a different objective.

I think the main message is now clearer and reach a sufficient level for publication : the authors propose a new method, based on cell enzymatic disaggregation, fluorescence microscopy and deep learning classification, to sort and identify cell types, as an alternative to multiplex stainings and would be of interest for the community. Sharing of the code and of the data set, as they now propose, will help other researchers to reproduce the experiment. Extra sharing of the JOB NIS protocol for automation (exposure time etc...) would be even nicer.

Please find enclosed our revised manuscript, “Image-based Phenotyping of Disaggregated Cells Using Deep Learning,” for publication in Communications Biology.

We have addressed the minor comments from the reviewers and the publishing requirements. We would like to thank you and the reviewers for guiding this article to its final form.

Reviewer #1:

Item 5: Thank you for tempering the language and adding more example images. However, I still think the claim they are indistinguishable by eye is too strong. Looking at S1 to S8, I agree some are very similar, but some are “easy” to differentiate even for me (not a trained expert in the field). For example, S1 seems distinctive, as do S7 and S8. Please could the authors further re word the claims, or better justify them.

We thank the reviewer for this comment and have tempered line 23 as follows. We hope the move from “indistinguishable” to “difficult” tempers the statement to align with the reviewer’s view.

Machine learning has been used for image cytometry but has been limited by cell agglomeration and it is currently unclear if this approach can reliably phenotype cells that are difficult to distinguish by the human eye.

Minor comments:

L359 thresholded*

L237 “trained cell lines *and?* were generally”

Reviewer #2:

One comment for reuse is that the true cell class seems to be given by the file name, but this could be specified in the text to help reproducing the results.

We appreciate this comment as it helps improve the reproducibility of our study. We have added a legend to the article cover letter, and further created a README file in the database.

Extra sharing of the JOB NIS protocol for automation (exposure time etc...) would be even nicer.

We thank the reviewer for bringing up this point. The JOB protocol has been uploaded to our Github repository at <https://github.com/SamBerryman/Image-based-Cell-Phenotyping-Using-Deep-Learning>.